# Chiroptical Spectroscopy, Theoretical Calculations, and Symmetry of a Chiral Transition Metal Complex with Low-Lying Electronic States

**DOI:** 10.3390/molecules30040804

**Published:** 2025-02-10

**Authors:** Mutasem Alshalalfeh, Yunjie Xu

**Affiliations:** Department of Chemistry, University of Alberta, Edmonton, AB T6G 2G2, Canada; alshalal@ualberta.ca

**Keywords:** vibrational circular dichroism, low-lying electronic excited states (LLESs), conformational search, symmetry consideration, DFT calculations, chiral transition metal complexes

## Abstract

Vibrational circular dichroism (VCD) enhancement by low-lying electronic states (LLESs) is a fascinating phenomenon, but accounting for it theoretically remains a challenge despite significant research efforts over the past 20 years. In this article, we synthesized two transition metal complexes using the tetradentate Schiff base ligands (*R*,*R*)- and (*S*,*S*)-*N*,*N*′-Bis(3,5-di-tert-butylsalicylidene)-1,2-cyclohexanediamine with Co(II) and Mn(III), referred to as Co(II)-salen-chxn and Mn(III)-Cl-salen-chxn, respectively. Their stereochemical properties were explored through a combined experimental chiroptical spectroscopic and theoretical approach, with a focus on Co(II)-salen-chxn. Extensive conformational searches in CDCl_3_ for both high- and low-spin states were carried out and the associated infrared (IR), VCD, ultraviolet-visible (UV-Vis) absorption, and electronic circular dichroism (ECD) spectra were simulated. A good agreement between experimental and simulated data was achieved for IR, VCD, UV-Vis, and ECD, except in the case of VCD of Co(II)-salen-chxn which exhibits significant intensity enhancement and monosignate VCD bands, attributed to the LLESs. Interestingly, detailed comparisons with Mn(III)-Cl-salen-chxn and previously reported Ni(II)-salen-chxn and Cu(II)-salen-chxn complexes suggest that the enhancement factor is predicted by the current density functional theory simulations. However, the monosignate signatures observed in the experimental Co(II) VCD spectrum were not captured theoretically. Based on the experiment and theoretical VCD and ECD comparison, it is tentatively suggested that Co(II)-salen-chxn exists in both low- and high-spin states, with the former being dominant, while Mn(III)-Cl-salen-chxn in the high-spin state. The study indicates that VCD enhancement by LLESs is at least partially captured by the existing theoretical simulation, while the symmetry consideration in vibronic coupling provides further insight into the mechanisms behind the VCD sign-flip.

## 1. Introduction

Schiff base ligands belong to a category of organic compounds distinguished by the presence of an imine (-C=N-) functional group, resulting from the reaction between the amine amino group and the aldehyde or ketone carbonyl group [1]. These ligands have gained significant attention due to their unique coordination chemistry properties. This special class of ligands, known as ‘privileged ligands’, can coordinate with a variety of metal ions and lanthanides, forming functional metal complexes in diverse oxidation states and geometries [2]. The unique properties of Schiff bases and their complex transition metal derivatives, mainly attributed to the imine functionality and its chelating character, have sparked the interest of researchers to explore their potential in a wide range of fields, such as enantioselective catalysis [3], chiral sensing [4], and biomedical applications [2]. Detailed knowledge of the stereochemical properties of Schiff base ligands and their metal complexes including ligand chirality, induced chiral configuration of the metal centers, and conformational distributions is essential to advance these important applications.

Electronic circular dichroism (ECD), vibrational circular dichroism (VCD), and Raman optical activity (ROA), especially in combination with density functional theory (DFT) calculations, have been effectively utilized to extract chirality-related information of these transition metal complexes and their ligands directly in solution [5,6,7,8,9,10]. Generally, VCD and ROA, characterized by many well-resolved vibrational bands, are particularly effective in providing decisive insights into chirality, conformational distributions, and intra- and intermolecular noncovalent interactions, both with solvents [11,12,13,14,15] and within solute itself [16,17]. Such detailed insights, however, are typically only achievable through the combination of DFT calculations [18,19,20].

One common challenge is that VCD (or ROA) band intensity is on the order of 10^−6^ to 10^−4^ (10^−5^ to 10^−3^) relative to the parent IR (Raman) bands, thus requiring a long acquisition time as well as a high sample concentration for reliable VCD (ROA) measurements. Therefore, much research efforts have been devoted to VCD [21,22,23] and ROA [24] intensity enhancement, as well as the mechanisms of chirality transfer [25] and enhancement [26,27,28,29]. For example, strongly induced solvent chiral Raman signals in solutions with a chiral Ni(II) transition metal complex were also reported, leading to the very recent discovery of a new type of chiral Raman spectroscopy named ECD-CP-Raman, abbreviated as eCP-Raman [30]. By using a metal ion bound to a ligand, one can greatly enhance the typically weaker ligand ROA and VCD intensities, as reported in several Co(II) [31], Ni(II) [32,33], and Cu(II) [34,35] complexes. For example, Domingos et al. showed that adding a Cobalt salt can drastically amplify VCD signals associated with the local chiral environment in biomolecules [31]. Later, Bürgi and co-workers reported a significant amplification of VCD signals of a thiolate-protected gold cluster Au_25_(Capt)_18_ (Capt = captopril) in aqueous solution by simply adding a cobalt salt to the aqueous solution [36]. More recently, a comprehensive review highlights many examples, including lanthanide complexes, supramolecular assemblies, amyloid fibrils, and protein or peptide aggregates, that exhibit enhanced VCD intensities [21].

Nafie’s single electronic state approximation offers a theoretical framework for understanding how low-lying electronic states (LLESs) contribute to VCD enhancement [27,37]. However, current DFT calculations have yet to successfully replicate the corresponding experimental VCD enhancement results. To address this limitation, research groups have developed approximation methods [38] and tools for visualizing vibrational transition current density [39] to aid these efforts. For example, Tomeček and Bouř introduced a method to account for the coupling between the LLESs and vibrational states to reproduce the enhanced VCD signals observed experimentally [38].

In this paper, we applied UV-Vis, ECD, IR, and VCD spectroscopies to investigate a synthesized chiral transition metal complex, namely *N*,*N*′-Bis(3,5-di-tert-butylsalicylidene)-1,2-cyclohexanediaminocobalt(II) (Co(II)-salen-chxn) (see Figure 1), as well as a related Mn(III)-Cl-salen-chxn complex (see Appendix A) for comparison. While the experimental IR and VCD spectra of these two complexes had been reported previously [40], this study focused on extensive theoretical calculations for potential conformers of the complex with Co in various oxidation and spin states, along with the corresponding calculations for the Mn complex and symmetry consideration. Simulated UV-Vis, ECD, IR, and VCD spectra were compared with the experimental data to determine the oxidation and spin state of the Co complex. Importantly, we investigated the factors contributing to agreement or disagreement between the experimental and simulated VCD bands, with particular attention to the recently proposed symmetry requirement [5,6] for coupling between the vibrational states and LLESs. This study offers new insights into the mechanisms responsible for VCD intensity enhancement and sign reversal.

We note that although serious attempts were made to obtain experimental Raman and ROA spectra of the named complexes, the measurements were not successful because of severe fluorescence interference and are therefore not included in the current paper.

## 2. Results and Discussion

In Section 2.1, we compared the experimental UV-Vis/ECD and IR/VCD spectra of the salen-chxn ligand, the Co(II) and Mn(III) complexes, and two previously reported metal-salen-chxn complexes [41]. Section 2.2 described the conformational searches and DFT geometry optimizations performed for the two metal complexes, and summarized all the low-energy minima identified in CDCl_3_. In Section 2.3, we compared the simulated UV-Vis/ECD spectra for both low- and high-spin states with the experimental spectra of the Co(II)-salen-chxn complex to extract information about its spin state. Section 2.4 presented comparisons and analyses of the experimental and theoretical IR and VCD spectra of the Co(II)-salen-chxn and Mn(III)-Cl-salen-chxn complexes in CDCl_3_. Discrepancies between experiment and theory highlight the influence of LLESs on the Co(II) VCD spectrum. Finally, we examined the C_2_ symmetry of the Co(II)-salen-chxn complex, assigned symmetry labels for the relevant vibrational modes and LLESs, and used the results to justify which vibrational bands could or could not couple with LLESs. This analysis provides insight into the VCD sign reversal observed in specific bands.

### 2.1. Comparison of the Experimental UV-Vis, ECD, IR, and VCD Spectra of the Co(II)-Salen-Chxn Complex with Those of Its Ligand and Other Related Complexes

The experimental UV-Vis and ECD spectra of the ligand salen-chxn and its metal complexes (Co(II), Mn(III), Cu(II), and Ni(II)) are presented in Figure 1. The UV-Vis spectrum of the ligand salen-chxn exhibits three main broad bands at 225 nm, 260 nm, and 325 nm, corresponding to the π−π* transitions of the ligand’s aromatic chromophores [41]. In the UV-Vis spectra of the metal complexes, the absorption band features extend to longer wavelengths (up to 500 nm), with new bands attributed to electronic transitions arising from the metal interactions with the ligand’s O and N atoms. The ECD spectrum of the salen-chxn ligand also differs notably from those of the metal counterparts. For example, in the 350–500 nm region, the ligand shows no visible ECD features, whereas each metal complex exhibits distinctive ECD bands in this range.

Figure 2 shows the experimental IR and VCD spectra of the ligand salen-chxn and its metal-salen-chxn complexes. The IR spectra of the pure ligand and the metal complexes all exhibit a strong band around 1600 cm^−^^1^ and similar features in the 1150–1300 cm^−^^1^ region. However, noticeable differences appear in the 1300–1550 cm^−^^1^ region, where the metal complexes show distinct features compared to the ligand. Interestingly, the observed IR band patterns of different metal complexes remain largely consistent. The presence of metal centers with varying oxidation states and spin multiplicities appears to have no substantial impact on the IR spectra.

Conversely, the VCD spectra of the metal-salen-chxn complexes differ drastically from that of the pure ligand and from each other, as illustrated in the lower panel of Figure 2. One exception is the considerable similarity between the VCD features of Ni(II) and Mn(III). Furthermore, it is noteworthy that the VCD spectra of the Cu(II), Ni(II), and Mn(III) complexes display both positive and negative bands, whereas the VCD spectrum of Co(II)-salen-chxn in this range consists almost entirely of monosignate bands with significantly enhanced g-factors. For instance, the band at approximately 1619 cm^−^^1^, corresponding to the C=N symmetric stretch, exhibits g-values of 2.1 × 10^−^^3^, 1.0 × 10^−^^4^, 1.3 × 10^−^^4^, and 1.8 × 10^−^^4^ for the Co(II), Mn(III), Cu(II), and Ni(II) complexes, respectively. Similarly, the g-factors for the aromatic ring C=C stretches at around 1526 cm^−^^1^ are 2.9 × 10^−^^3^, 0.6 × 10^−^^4^, 0.5 × 10^−4^, and 1.4 × 10^−^^4^ for the Co(II), Mn(III), Cu(II), and Ni(II) complexes, respectively.

To confirm the monosignate phenomenon for the Co(II)-salen-chxn complex, the experimental A and ∆A spectra of enantiomeric pairs were measured in the range 1400 to 3800 cm^−1^, as summarized in Figure 3. The IR spectra of the enantiomers show nearly identical features as expected, while the ΔA spectra exhibit clear mirror image quality. Interestingly, the ΔA features contain a broad, monosignate feature spanning from 1700 to 3800 cm^−1^. This unusual broad feature corresponds to an ECD band centered around 2300 cm^−1^, arising from LLES transitions within the Co(II)-salen-chxn complex. Similar broad ECD features have been reported for other Schiff base Co(II) complexes, such as an ECD band centered around 2700 cm^−1^ for Co(II)-sparteine [40] and at 2100 cm^−1^ for Co(II)-saldiphenyl [42,43]. This assignment is further supported by the TDDFT calculations described in Section 2.3. Additionally, weaker, sharp bands appear in the ΔA spectrum between 2800 and 3100 cm^−1^, corresponding to the C-H stretching vibrations of the complex. These VCD bands also show good mirror imaged quality for the enantiomeric pair. It is, however, difficult to extract the g-values of these broad ECD bands since the corresponding LLES transitions are very flat in the related IR spectra.

The following sections present a detailed theoretical exploration and symmetry analysis of the vibronic coupling to explain the observed enhancement phenomenon.

### 2.2. The Low-Energy Conformers of the Co(II)-Salen-Chxn Complex

It is well known that chiroptical spectroscopic features, especially those associated with VCD and ROA, are highly sensitive to not only chirality but also conformations of chiral molecules. Moreover, the oxidation states and spin multiplicities of the metal center may also play significant roles in the appearance of VCD and ROA spectra. We therefore set out to evaluate the influence of conformational distributions, associated oxidation states, and spin multiplicities of the metal center on ECD and VCD spectra.

For the cobalt compound, a single mass spectrum peak at ~603.3 (as shown in Appendix A) was observed, consistent with the Co-salen-chxn with a +1 charge, i.e., Co(III)-salen-chxn. Since the mass spectrum cannot tell us whether the species in solution is Co(II) which may lose one electron in the electrospray ionization process, or if the original species is Co(III) in solution, both oxidation states were considered. For a square planar Co(II) complex with an electronic configuration of d^7^, the high-spin state has a spin multiplicity of S = 4, while the low-spin state has a spin multiplicity of S = 2. Both spin states were considered below.

To systematically explore the possible conformers of the Co(II)-salen-chxn complex in CDCl_3_, we utilized the conformer–rotamer sampling tool (CREST) [44,45] by Grimme and co-workers. Built upon the previous semiempirical tight-binding (TB) quantum chemistry method, called GFN-xTB [46], the CREST code provides an efficient and accurate exploration and screening of the conformational space for molecules consisting of up to a thousand atoms in size. To include the bulk solvent environment in our searches, we applied the generalized Born (GB) model augmented with solvent-accessible surface area (SA), i.e., the GBSA implicit solvation model [47,48], using CDCl_3_ as the solvent. The power of CREST has been demonstrated by high resolution rotational spectroscopic studies of conformers of noncovalently bound clusters of a wide range of molecules [49,50,51] It has also been successfully used in solution VCD studies [52].

A large number of CREST geometry candidates of the salen-chxn ligand were previously reported [41]. These are associated with the cyclohexane ring conformations, the axial and equatorial positions of the two large substituents on the cyclohexane ring, the rotatable motions about the N-C_cyclohexane_ bonds, the orientation of the OH group, and the staggered and eclipsed conformations of the two tBu groups in each substituent.

The corresponding cobalt complex is considerably more rigid than the ligand due to the coordination bonds. As mentioned before, it was initially unclear if Co takes on Co(III) or Co(II) oxidation states. Therefore, CREST searches were carried out for both oxidation states. Twelve CREST candidates were identified for each oxidation state. The geometry optimization of all possible candidates at the B3LYP-D3BJ/6-311++G(d,p) level of theory revealed only three stable conformers of the complex. These conformers are associated with different tBu orientations. It is also interesting to note that the same conformers were detected for the Co(II) and Co(III)-salen-chxn complex after the final DFT geometry optimization, regardless of spin states. This suggests that using different oxidation states in the CREST searches does not impact the generation of potential conformers. However, different oxidation and spin states significantly alter the simulated IR and VCD spectral features, as discussed in the next section. For conciseness, only the optimized geometries of the three conformers for the Co(II)-salen-chxn complex in the high- and low-spin states, are summarized in Figure 4, together with their relative free energies and Boltzmann factors at 298 K. For comparison, the four most stable salen-chxn ligand conformers, optimized at the B3LYP-D3BJ/6-311++G(d,p) within an energy window of 15 kJ mol^−1^, are also included in Figure 4.

For a square planar Mn(III) complex with a d^4^ configuration, the high-spin state has four unpaired electrons, resulting in a spin multiplicity of S = 2 × 2 + 1 = 5. In the low-spin state, there are no unpaired electrons, giving a spin multiplicity of S = 2 × 0 + 1 = 1. The same theoretical procedure was applied to the Mn(III)-Cl-salen-chxn complex, resulting in the identification of thirteen CREST candidates. Geometry optimization of all possible candidates at the B3LYP-D3BJ/6-311++G(d,p) level of theory revealed four stable conformers of the Mn(III)-Cl complex together with their relative free energies at both high- and low-spin states and their Boltzmann factors at 298 K, as shown in Appendix A.

Another crucial aspect to consider is the absolute configuration of the chiral metal center, specifically the helicity at the metal center. Given the nearly square geometries displayed by all these metal complexes, it is conceivable that both Λ (or M in some publications) and Δ (or P) helicity are achievable. A slight alteration in the dihedral angle of O-Metal-N in the opposite direction would be sufficient to switch helicities. With ligands with *(R*,*R*) chirality, a distinct preference toward Λ metal chirality over Δ was observed in several previously reported complexes, such as the Co complex with (bis[(*R*/*S*)-N-(1-(Ar)ethyl)salicylaldiminato]) [6]. In the study of Schiff base ligand (*R*,*R*) and (*S*,*S*)-bis(pyrrol-2-ylmethyleneamine)-cyclohexane and its four mononuclear complexes with Ni(II), Cu(II), Pd(II), and Pt(II), only Λ metal chirality was observed, due to the strong constraint imposed by the cyclohexane ring [53]. Similarly, in the recent investigation into the metal chirality of Ni(II) and Cu(II) with the same salen-chxn ligand, no Δ helicity geometry candidates were identified [41].

### 2.3. Simulated UV-Vis and ECD Spectra and the Experimental Results

Since the simulated IR spectra of all Co(III) conformers do not match the experimental IR spectrum (see Section 2.4), only the theoretical results for Co(II) will be presented in the remainder of the paper, unless stated otherwise.

According to crystal field theory, square planar complexes are typically low-spin due to their high crystal field splitting energy. However, high-spin Co(II) ions in square planar coordination have also been reported [54,55]. It would be intriguing to investigate whether high-spin and low-spin configurations produce any detectable differences in UV-Vis and ECD spectra. In Figure 5, we present a comparison of the simulated average UV-Vis and ECD spectra for both low-spin and high-spin configurations with the experimental spectra. Additionally, the simulated, individual UV-Vis and ECD spectra of the dominant conformers of the Co(II)-salen-chxn complex in low- and high-spin states in acetonitrile are shown in Appendix A, respectively.

The UV-Vis and ECD spectra of Co(II)-salen-chxn conformers I, II, and III in low-spin display a remarkable degree of similarity. This similarity can be attributed to the UV-Vis and ECD features being relatively insensitive to subtle variations in tBu conformations. A similar observation applies to the high-spin state. Moreover, the Boltzmann-averaged UV-Vis spectra for both high-spin and low-spin states are largely similar, with only minor differences in the 200–700 nm. In contrast, the Boltzmann-averaged ECD spectra of the high- and low-spin states show significant differences within the same range (see Figure 5), highlighting a clear distinction between the two states. Although the broad nature of the spectral features in the UV-Vis region made conclusive assignment challenging, the simulated high-spin ECD spectrum aligns better with the experimental data than the low-spin spectrum, especially in the 200–300 nm range, successfully reproducing the key experimental features.

Another goal is to understand the origin of the broad positive or negative spectral band observed for the (*R*,*R*) or (*S*,*S*) enantiomer in the experimental VCD spectrum in the 1700–3800 cm^−1^ region (Figure 3), respectively. To verify if this observation is due to the d-d LLES transitions of Co(II)-salen-chxn, the averaged theoretical ECD spectra of the high-spin and low-spin states in this targeted range were simulated, and compared with the experimental chiroptical spectrum in Figure 6.

In the experimental spectrum, the ECD band is consistently positive in the region from 1600 to 3800 cm^−1^, with the band maximum located around 2400 cm^−1^. To facilitate the comparison with the experimental data, the ECD band centers are red-shifted by about 700 cm^−1^. A similar red-shift of approximately 1000 cm^−1^ was previously reported for TDDFT calculations, as well as for more sophisticated approaches, such as adiabatic Hessian calculations [56]. The simulated low-spin ECD spectrum displays a broad band with only a positive sign, aligning well with the experimental features, whereas the simulated high-spin ECD spectrum features a small positive band and a strongly negative band, inconsistent with the experimental data. Overall, the good agreement between the experimental and simulated UV-Vis and ECD spectra of the low-spin state in both the 200–500 nm and in the 1600–3800 cm^−1^ regions suggests that low-spin Co(II) predominates in solution. It is recognized that a small contribution of the high-spin species may further improve the agreement between the experimental and theoretical data shown in Figure 6, a point which will be further addressed in Section 2.5. Based on the TDDFT calculations and the experimental data, it can be concluded that the broad chiroptical bands observed in the IR region are due to the ECD bands of the LLES transitions.

### 2.4. Comparison of the Experimental and Simulated IR and VCD Spectra

With regard to whether the Co complex exists in the oxidation states of Co(III) or Co(II), we compared the simulated IR spectra of all conformers of Co(III)-salen-chxn (Appendix A). Clearly, the simulated IR spectra of Co(III) disagree with the experimental IR spectrum. For conciseness, only the calculation results of Co(II)-salen-chxn will be described in the remainder of the paper.

As mentioned previously, it is of considerable interest to investigate whether the low- and high-spin configurations of Co(II) give rise to any noticeable differences in their IR and, especially, VCD features. The individual IR and VCD spectra of the three most stable low-spin Co(II)-salen-chxn conformers are depicted in Figure 7a. The simulated IR and VCD spectra of the individual conformers are almost the same, except in the 1440–1470 cm^−1^ and 1150–1225 cm^−1^ regions (highlighted with green boxes). For example, in the 1440–1470 cm^−1^ region, primarily corresponding to the bending motions of the C-H of the tBu groups, conformer I exhibits a negative-positive pattern from low to high cm^−1^. In contrast, conformers II and III display opposite sign patterns, i.e., positive–negative, after amplification. Moreover, the band at the 1150–1225 cm^−1^ region, which is assigned to the C-H wagging motion from the tBu groups, conformers II, and III show a negative VCD band, whereas this particular VCD band is absent for conformer I.

Similarly, the IR and VCD spectra of the high-spin Co(II)-salen-chxn conformers were simulated at the same level of theory. The individual conformer IR and VCD spectra of the three most stable high-spin Co(II)-salen-chxn conformers are provided in Figure 7b. Again, individual IR spectra of the Co(II)-salen-chxn conformers at high-spin look almost identical, whereas the corresponding VCD spectra exhibit minor differences, highlighted in two green boxes. For example, conformer I has a negative VCD band at 1347 cm^−1^, which is associated with the C-H cyclohexane wagging motion; conversely, conformers II and III display a very small positive band at about the same position. A strong negative band attributed to the C=C stretching from the benzene ring was predicted in the VCD spectra of conformer III at 1335 cm^−1^, whereas the corresponding VCD band is not visible for conformers I and II. Additionally, for conformer III, the C-H twisting motions of the cyclohexane and benzene ring at about 1143 cm^−1^ exhibit a much different pattern in terms of sign and relative intensity when compared to those of conformers I and II. Overall, the different tBu conformations appear to have a minimum effect on the IR and VCD spectra of both high- and low-spin Co(II)-salen-chxn.

Last, the experimental IR and VCD spectra in the 950–1700 cm^−1^ range are compared with the corresponding Boltzmann-averaged spectra of the low- and high-spin Co(II)-salen-chxn in Figure 8. The simulated IR spectra of low- and high-spin species look quite similar, except that the high-spin species exhibits significantly broader (unresolved) band features around 1600 cm^−1^ in contrast to the sharper feature of the low-spin species. Additionally, some minor differences in the relative intensities and patterns of some bands could be detected in the 950–1050 cm^−1^ region and around 1360 cm^−1^.

In the optimized high-spin and low-spin Co(II)-salen-chxn geometries, the metal-ligand bond lengths show similar differences for these two spin states to those reported in the literature [57,58]. In the low-spin state, the Co-N and Co-O bond lengths are 1.886 Å and 1.877 Å, respectively, whereas in the high-spin state, the bond lengths are longer, with Co-N and Co-O bond lengths of 2.038 Å and 1.942 Å, respectively. As a result, some minor differences were observed in the IR spectrum of the low-spin state compared to that of the high-spin state, as discussed before [57]. Nevertheless, both high- and low-spin IR spectra show good agreement with the experimental IR spectrum.

In contrast, the Boltzmann-averaged VCD spectra of the high- and low-spin Co(II)-salen-chxn exhibit noticeable differences. For instance, two main VCD bands in the region below 1200 cm^−1^ are of opposite signs for high- and low-spin. Moreover, the simulated VCD spectrum of the high-spin state has a negative band at ~1623 cm^−1^, which is completely absent in the simulated low-spin VCD spectrum. Furthermore, the low-spin VCD bands are considerably more intense than the associated ones in the high-spin VCD spectrum in the 1500–1700 cm^−1^, while their corresponding IR bands are of similar intensity. Clearly, in comparison with the experimental VCD spectrum, neither the high-spin nor low-spin simulations were able to reproduce the experimental monosignate character, an important point which will be further discussed.

With regards to the open-shell Mn(III)-Cl-salen-chxn complex, the simulated IR and VCD spectra for the high-spin configuration are provided in Figure 9, together with the experimental data which exhibits no obvious VCD enhancement. Overall, the simulated spectra closely match the experimental observations and all medium to strong positive and negative VCD bands are well reproduced. Additionally, the experimental g-factors (IR/VCD intensity ratio) are well reproduced in the simulated spectra. For example, the experimental g-factors for the strong bands located at 1619 cm^−1^ (assigned to the C=N symmetric stretch) and 1526 cm^−1^ (aromatic ring C=C stretches) are 1 × 10^−4^ and 0.6 × 10^−4^, respectively. These values were correctly predicted in the high-spin simulated spectra as 0.6 × 10^−4^ and 0.4 × 10^−4^. The simulated IR and VCD spectra for the low-spin configuration, in contrast, agree poorly with the experimental results, as illustrated in Appendix A. These comparison results indicate that the Mn(III)-Cl-salen-chxn complex exists as a high-spin species in CDCl_3_.

While VCD simulations generally align well with experimental observations for closed-shell transition metal complexes, such as the Ni(II)-salen-chxn complex [41], simulating open-shell transition metal complexes is more complicated and less predictable. For example, Cu(II)-salen-chxn, an open-shell complex with 3d^9^ configuration, shows no VCD intensity enhancement (Figure 2) Similarly, high-spin Mn(III)-Cl-salen-chxn with d^4^ configuration exhibits no obvious VCD intensity enhancement (Figure 9). Furthermore, unlike Co(II)-salen-chxn whose VCD is monosignate, Cu(II)-salen-chxn [41] and Mn(III)-salen-chxn display both positive and negative VCD bands, and their VCD spectra are well reproduced using the current VCD theoretical methodology.

As established in Section 2.1 and Section 2.3, Co(II)-salen-chxn has LLES d-d transitions. Based on the work by Nafie and co-workers [27,32], the presence of the LLESs is responsible for the enhancement. As pointed out in [27], the contribution to the atomic axial tensor (ATT) can be classified into three terms: (1) the regular term for systems without LLES, (2) the first correction term for systems with LLESs which depends only on the electronic energies in the usual way and can be calculated using standard VCD algorithms, and (3) the second correction term which depends on vibronic coupling between the LLESs and vibrational modes. It is interesting to note that even without the vibronic coupling correction term, the g-factors for the two strong VCD bands of low-spin Co(II)-salen-chxn at 1602 and 1520 cm^−1^ are predicted to be 2.0 × 10^−3^ and 2.0 × 10^−3^, respectively, comparable to the experimental values of 2.1 × 10^−3^ (1601 cm^−1^) and 2.9 × 10^−3^ (1520 cm^−1^). The corresponding high-spin g-factors for these two bands are 1.2 × 10^−3^ (1610 cm^−1^) and 1.0 × 10^−3^ (1520 cm^−1^). It appears that the significant VCD intensity enhancement is reproduced by the existing VCD simulation algorithm. On the other hand, the experimental monosignate VCD bands, a hallmark of vibronic coupling, are not reproduced theoretically. This is unsurprising since the current theoretical VCD approach does not take into account the aforementioned vibronic coupling. Indeed, similar observations have been well documented in the literature for other chiral transition metal complexes with LLESs [5,6].

Recently, Tomeček and Bouř introduced a novel perturbation approach, referred to as the beyond the Born-Oppenheimer (BBO) method, to address the vibronic coupling challenge [38]. Please refer to the introduction of [38] for other related, alternative theoretical method developments. In this approach, the coupling between the electronic and vibrational states is estimated using the harmonic approximation and simplified wavefunctions extracted from DFT calculations. The authors applied the BBO approach to calculate the VCD spectrum of the salicyl−Co(II) complex which has LLESs. Encouragingly, the BBO approach generated simulated VCD spectra featuring significantly enhanced VCD bands, bringing the predicted VCD intensities into the same range as the experimental values although the monosignate nature of the salicyl−Co(II) VCD spectrum remained elusive. Based on their study, reproducing the monosignate VCD spectrum of Co(II)-salen-chxn will likely remain challenging, even with the recent advanced BBO treatment.

Despite the challenges in accurately capturing the LLES-induced peculiar monosignate VCD features, the simulated VCD spectra for both high-spin and low-spin states seem to correctly predict the strong VCD features in the range of 1500 to 1700 cm^−1^. One might question whether this observation is coincidental or has underlying scientific significance. Recent studies [6,7] proposed that experimental monosignate VCD features could be analyzed by considering the symmetry requirement for vibronic coupling between vibrational normal modes and LLESs. This proposition motivated us to perform related analyses, which are described in the next section.

### 2.5. Discussion on Symmetry-Dependent VCD Enhancement

Since the VCD features are very much the same for the three Co(II)-salen-chxn conformers which differ in only their tBu internal rotation configurations, we focused on the most stable conformer for conciseness. This nearly planar coordination conformer is of C_2_-symmetry with A and B symmetry species. In Figure 10, two examples of vibrational normal modes with A and B symmetries of the high-spin species are presented.

In Figure 11, we labeled the prominent VCD bands of the low-spin and high-spin species with A and B symmetries. Some bands have both labels because there are multiple vibrational modes under one visible band. In the analysis of salicyl−Co(II), a pseudo-tetrahedral coordination complex with C_2_-symmetry, the authors verified that the first three LLESs are of B symmetry label [6]. Following the same procedure, we identified that the first three LLESs are of B symmetry in the case of Co(II)-salen-chxn. It is interesting to point out that the simulated *isolated* VCD bands with the A-symmetry generally align with the corresponding experimental VCD features in terms of signs and relative intensities, particularly below 1300 cm^−1^ and above 1400 cm^−1^. The 1300–1400 cm^−1^ region is less clear due to band crowding. Those *isolated* VCD bands labeled with the B symmetry are negative and are in disagreement with the monosignate positive experimental VCD bands. The observation is consistent with the assumption that only vibrational modes with the same symmetry species as that of the LLESs can participate in vibronic coupling and borrow magnetic dipole moment from the LLESs. Furthermore, the current comparison suggests that the VCD patterns of the normal vibrational modes with the A-symmetry can be generally properly modeled with the current DFT calculations in the present case.

Assuming that the VCD features for the A-symmetry vibrational modes are correctly sign-predicted for the high-spin and low-spin species, it is particularly interesting to look at the negative VCD band at about 1621 cm^−1^. The high-spin species exhibit an obvious negative VCD band at approximately 1621 cm^−1^, consistent with the experimental observation. On the other hand, a similar feature at this wavenumber is noticeably absent in the low-spin species. This suggests that the Co(II)-salen-chxn complex predominantly adopts a low-spin state with some contribution from the high-spin state in solution, which aligns with the conclusion drawn from the ECD comparison discussed in Section 2.3 (see Figure 5 and Figure 6).

## 3. Materials and Methods

### 3.1. Experimental

The (*R*,*R*) and (*S*,*S*) salen-chxn ligands (98%) were purchased from Sigma-Aldrich and used without further purification. The chiral transition metal complex Co(II)-salen-chxn was synthesized based on the previous literature procedures [59]. Briefly, the salen-chxn ligand was dissolved in anhydrous ethanol, and then the solution was heated to reflux until boiling. Cobalt (II) perchlorate hexahydrate in absolute ethanol was then added to the solution and refluxed for 3 h. Next, the solution mixture was concentrated using a rotavapor, and the residue was further dissolved in dichloromethane and ethyl acetate. The resulting solution was then filtered under suction, and the solvent in the filtrate was removed using a rotavapor to collect the remaining solid powder, which was further recrystallized using a mixture of dichloromethane and ethanol. The mass spectrum of the synthetic sample revealed only one major peak at 603.3354 (Appendix A), consistent with the mass expected based on the formula given in Figure 1, assuming a charge of +1. The same procedure was applied for the preparation of Mn(III)-Cl-salen-chxn (structural formula shown in Appendix A), and manganese(III) acetate dihydrate was used as the metal source.

The experimental IR and VCD spectra of Co(II)-salen-chxn and Mn(III)-Cl-salen-chxn were collected using an FTIR spectrometer (Bruker Vertex 70, Milton, ON, Canada) coupled to a VCD model (PMA 50). The deuterated chloroform was used as a solvent to prepare a solution with a concentration of 60 mg/1 ml. A demountable BaF_2_ window cell was used for all measurements with a 50 μm spacer. The PEM (photoelastic modulator) was set at 1400 cm^−1^ for the VCD spectra in the range 950–1700 cm^−1^, and it was set at 3000 cm^−1^ for VCD spectra in the range 1700–3800 cm^−1^. The collection time is 2 h. The same procedure was used for the measurements of Mn (III)-salen-chxn.

The UV-Vis and ECD spectra of the ligand salen-chxn and the Co (II) transition metal complex in acetonitrile were measured using a Jasco-1700 spectrometer with a variant concentration of 0.2 mM and a path length of 1 cm.

### 3.2. Theoretical

To systematically explore the possible conformers of the Co(II)-salen-chxn complex in CDCl_3_, we applied the CREST [44,45] program with GBSA implicit solvation model [47,48] and with the GFN2-xTB [60] level of theory. To reduce computational costs, we implemented a multitiered approach [61]: (i) apply CREST to generate initial geometry candidates; (ii) perform a relaxed geometry optimization at the revPBE-D3/def2-SVP level [62], with the empirical D3 dispersion correction [63] for the CREST candidates, followed by a single-point energy evaluation at the B3LYP-D3/def2-TZVP level of the optimized structures; and (iii) Gaussian 16 package [64] was used to perform the final geometry optimization and harmonic frequency calculations of all possible conformers. In the current study, the conformational search of the Co(II)-salen-chxn complex was carried out at different oxidation states and different spins. The B3LYP-D3BJ/6-311++G(d,p) [65] level of theory was used for the final geometry optimization and IR/VCD calculations, where BJ stands for the Becke-Johnson damping function [66]. The implicit solvent was included using the integral equation formalism (IEF) version of the PCM [67] to account for the bulk solvent environment.

The time-dependent DFT (TDDFT) calculations were employed to calculate the excited state energies and oscillator strengths. In general, it is challenging to accurately capture excited state properties, especially for open-shell systems, and the performance of different combinations of DFT functionals and basis sets can vary considerably. Based on the previous ECD simulations of Co(II) transition metal complexes [5,68] and of other related Ni(II) and Cu(II) complexes [41], we chose the combination of B3LYP-D3BJ/6-311++G(d,p) level of theory with the PCM of acetonitrile. The first 250 electronic states were considered in the simulation of the UV-Vis and ECD spectra for the Co(II)-salen-chxn complex, using a Gaussian line shape with a half-width at half-height (HWHH) of 0.15 eV.

## 4. Conclusions

The stereochemistry properties of two open-shell Co(II)-salen-chxn and Mn(III)-Cl-salen-chxn complexes were investigated using the UV-Vis, ECD, IR, and VCD chiroptical tools, in tandem with theoretical modeling. While the former demonstrates strong VCD intensity enhancement and monosignate character, the latter exhibits regular VCD behavior. Systematic conformational searches were carried out for the Co(II) complex with different metal oxidation states and spin states. Comparison between the experimental and theoretical IR spectra confirmed that Co(II) is the proper oxidation state. The three low-energy conformers of Co(II)-salen-chxn are associated with the different tBu arrangements. The UV-Vis, ECD, IR, and VCD spectra of these three conformers with two different spin states were calculated at the B3LYP-D3BJ/6-311++G(d,p)/PCM (CHCl_3_) level of theory. The comparison between the experimental and theoretical UV-Vis, ECD, IR, and VCD spectra led to the conclusion that the Co(II)-salen-chxn complex may coexist in low-spin and high-spin states, dominated by the former, whereas the Mn(III)-Cl-salen-chxn complex exists in high-spin. While the experimental VCD features of Mn(III)-Cl-salen-chxn were well reproduced theoretically, the monosignate VCD character of the Co(II)-salen-chxn was not captured by the current simulation. This disagreement is attributed to strong vibronic coupling in Co(II)-salen-chxn which has LLESs. By using the symmetry selective vibronic coupling consideration, the strong and well-resolved VCD bands with A-symmetry labels were correctly predicted, offering insights into the experimental VCD features of Co(II)-salen-chxn. We hope that the current work will generate interest in future theoretical development in treating the LLES effect in VCD.

## Data Availability

Data are contained within the article.

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
