# Peer review of "Chiroptical Spectroscopy, Theoretical Calculations, and Symmetry of a Chiral Transition Metal Complex with Low-Lying Electronic States"

_molecules, 2025, doi:10.3390/molecules30040804_

Round 1

Reviewer 1 Report

Comments and Suggestions for Authors

The manuscript investigates the enhancement of vibrational circular dichroism (VCD) by low-lying electronic states (LLESs) through experimental and theoretical studies of Co(II)-salen-chxn and Mn(III)-Cl-salen-chxn complexes. This highly relevant topic addresses a long-standing challenge in chiroptical spectroscopy. By combining comprehensive experimental data (VCD, IR, UV-Vis, and ECD) with density functional theory (DFT) simulations, the study provides valuable insights into spin-state contributions and the mechanisms underlying VCD enhancement. I recommend the manuscript for publication in its current form.

The main question addressed by the research is how low-lying electronic states (LLESs) enhance vibrational circular dichroism (VCD) and whether this enhancement can be captured theoretically. The study aims to investigate the effects of spin-state contributions on VCD and the challenges in predicting these enhancements using density functional theory (DFT) simulations.

The topic is both original and highly relevant to the field. The enhancement of VCD by low-lying electronic states (LLESs) is a fascinating and challenging phenomenon that has been difficult to model theoretically, despite significant research efforts over the past 20 years. This research addresses the gap in understanding the mechanisms behind VCD enhancement, specifically the influence of spin-state configurations and vibronic coupling.

The study provides valuable insights into the spin-state contributions to VCD enhancement and how these contributions can be incorporated into theoretical models. By using transition metal complexes and combining both experimental and theoretical approaches, the authors provide a more nuanced understanding of the VCD enhancement mechanism compared to previous studies that primarily focused on simpler systems. The paper also addresses the theoretical difficulties in capturing monosignate VCD bands, which were observed experimentally but not fully predicted by simulations.

The authors could consider improving the theoretical models to better capture the monosignate VCD bands observed experimentally for Co(II)-salen-chxn. Additionally, expanding on the symmetry considerations in vibronic coupling and providing a more detailed comparison of the theoretical simulations with experimental data for other transition metal complexes would improve the methodology. More direct comparisons between high- and low-spin states could also help clarify the relationship between spin states and VCD intensity enhancement.

Author Response

Reviewer 1

Comment 1: The manuscript investigates the enhancement of vibrational circular dichroism (VCD) by low-lying electronic states (LLESs) through experimental and theoretical studies of Co(II)-salen-chxn and Mn(III)-Cl-salen-chxn complexes. This highly relevant topic addresses a long-standing challenge in chiroptical spectroscopy. By combining comprehensive experimental data (VCD, IR, UV-Vis, and ECD) with density functional theory (DFT) simulations, the study provides valuable insights into spin-state contributions and the mechanisms underlying VCD enhancement. I recommend the manuscript for publication in its current form.

The main question addressed by the research is how low-lying electronic states (LLESs) enhance vibrational circular dichroism (VCD) and whether this enhancement can be captured theoretically. The study aims to investigate the effects of spin-state contributions on VCD and the challenges in predicting these enhancements using density functional theory (DFT) simulations.

The topic is both original and highly relevant to the field. The enhancement of VCD by low-lying electronic states (LLESs) is a fascinating and challenging phenomenon that has been difficult to model theoretically, despite significant research efforts over the past 20 years. This research addresses the gap in understanding the mechanisms behind VCD enhancement, specifically the influence of spin-state configurations and vibronic coupling.

The study provides valuable insights into the spin-state contributions to VCD enhancement and how these contributions can be incorporated into theoretical models. By using transition metal complexes and combining both experimental and theoretical approaches, the authors provide a more nuanced understanding of the VCD enhancement mechanism compared to previous studies that primarily focused on simpler systems. The paper also addresses the theoretical difficulties in capturing monosignate VCD bands, which were observed experimentally but not fully predicted by simulations.

The authors could consider improving the theoretical models to better capture the monosignate VCD bands observed experimentally for Co(II)-salen-chxn. Additionally, expanding on the symmetry considerations in vibronic coupling and providing a more detailed comparison of the theoretical simulations with experimental data for other transition metal complexes would improve the methodology. More direct comparisons between high- and low-spin states could also help clarify the relationship between spin states and VCD intensity enhancement.

Response1: 

We thank the reviewer for the kind words and positive comments. We agree with the reviewer that including symmetry considerations in vibronic coupling is important. The only previous reports of the symmetry consideration are cited in the current paper as Refs 5 and 6. To expand similar analyses to other open-shell systems would be a major undertaking, worthy of further publications. In the current study, we also include other open-shell transition metals without obvious VCD enhancement for comparison to gain further insights into the LLES enhancement mechanism.

To the best of our knowledge, no existing theoretical models can account for the monosignate nature of the enhanced VCD spectra. This is an area of active research.

In Figure 8, we compare the simulated IR and VCD spectra for high-spin and low-spin states with the experimental ones. The IR intensities for both states are similar, but the VCD intensity for the low-spin state is notably higher than that of the high-spin state, indicating a notably higher g factor for the low-spin state, in accord with the experimental g factor.

We have also corrected two types on lines 318 and 319. The sentences should read “Overall, the good agreements between the experimental and simulated UV-Vis and ECD spectra of the low-spin state in both the 200-500 nm and in the 1600-3800 cm-1 regions suggests that low-spin Co(II) predominates in solution. It is recognized that a small contribution of the high-spin species may further improve the agreement between the experimental and theoretical data shown in Figure 6,…”

Reviewer 2 Report

Comments and Suggestions for Authors

The manuscript by Alshalalfeh et al. shows a comprehensive and detailed study on the stereochemical properties of chiral transition metal complexes, specifically Co(II)-salen-chxn and Mn(III)-Cl-salen-chxn. The authors employ both extensive experimental chiroptical spectroscopic methods (UV-Vis, ECD, IR, VCD) and theoretical density functional theory (DFT) calculations to explore low-lying electronic states (LLESs) and their impact on vibrational circular dichroism (VCD) intensity enhancement. The study offers new insights into the role of symmetry in vibronic coupling and VCD sign flip. The manuscript is well-suited for publication in Molecules after clarifying a few points.

1.      Is B3LYP-D3BJ  benchmarked in the field? Provide additional rationale for the choice of this level of theory.

2.      Given the current DFT calculations were unable to reproduce the monosignate VCD bands, the authors discussed the beyond Born-Oppenheimer (BBO) approach (ref. 33). Are there any other alternative computational methods that consider vibronic coupling and could potentially aid the experimental interpretation here?

Author Response

Reviewer 2

Comment 1: 

The manuscript by Alshalalfeh et al. shows a comprehensive and detailed study on the stereochemical properties of chiral transition metal complexes, specifically Co(II)-salen-chxn and Mn(III)-Cl-salen-chxn. The authors employ both extensive experimental chiroptical spectroscopic methods (UV-Vis, ECD, IR, VCD) and theoretical density functional theory (DFT) calculations to explore low-lying electronic states (LLESs) and their impact on vibrational circular dichroism (VCD) intensity enhancement. The study offers new insights into the role of symmetry in vibronic coupling and VCD sign flip. The manuscript is well-suited for publication in Molecules after clarifying a few points.

Response 1: 

We thank the reviewer for the kind words and positive comments. We address the comments below point-by-point.

Comment 2: 

Is B3LYP-D3BJ  benchmarked in the field? Provide additional rationale for the choice of this level of theory.

Response 2: 

Several levels of theory were benchmarked in the study of the related Ni(II) and Cu(II)-salen-chxn transition metal complexes (Ref. 36), and the B3LYP-D3BJ /6-311++G(d,p) level of theory accurately reproduced the UV, ECD, IR and VCD results.  

Comment 3:

Given the current DFT calculations were unable to reproduce the monosignate VCD bands, the authors discussed the beyond Born-Oppenheimer (BBO) approach (ref. 33). Are there any other alternative computational methods that consider vibronic coupling and could potentially aid the experimental interpretation here?

Response 3:

This is an interesting question. The Ref. 33 on beyond Born-Oppenheimer (BBO) approach is a rich source of references for other alternative computational method developments. Since the differences in alternative theoretical approaches can be subtle and intricated, we refer readers to the introduction part of Ref. 33 for details and have added a sentence on line 457 “Please refer to the introduction of Ref. 33 for other related, alternative theoretical method developments.” To the best of our knowledge, we are not aware of any current approaches that can reproduce the monosignate VCD bands reported here or in Refs. 5 and 6.